# The Machinery of Exosomes: Biogenesis, Release, and Uptake

**DOI:** 10.3390/ijms24021337

**Published:** 2023-01-10

**Authors:** Sofia V. Krylova, Daorong Feng

**Affiliations:** 1Departments of Medicine, Albert Einstein College of Medicine, New York, NY 10461, USA; 2Molecular Pharmacology, Albert Einstein College of Medicine, New York, NY 10461, USA; 3Norman Fleischer Institute of Diabetes and Metabolism, Albert Einstein College of Medicine, New York, NY 10461, USA

**Keywords:** exosome biogenesis, cargo sorting, exosome secretion, exosome uptake

## Abstract

Exosomes are a subtype of membrane-contained vesicles 40–200 nm in diameter that are secreted by cells into their surroundings. By transporting proteins, lipids, mRNA, miRNA, lncRNA, and DNA, exosomes are able to perform such vital functions as maintaining cellular homeostasis, removing cellular debris, and facilitating intercellular and interorgan communication. Exosomes travel in all body fluids and deliver their molecular messages in autocrine, paracrine as well as endocrine manners. In recent years, there has been an increased interest in studying exosomes as diagnostic markers and therapeutic targets, since in many disease conditions this machinery becomes dysregulated or hijacked by pathological processes. Additionally, delivery of exosomes and exosomal miRNA has already been shown to improve systemic metabolism and inhibit progression of cancer development in mice. However, the subcellular machinery of exosomes, including their biogenesis, release and uptake, remains largely unknown. This review will bring molecular details of these processes up to date with the goal of expanding the knowledge basis for designing impactful exosome experiments in the future.

## 1. Exosome Biogenesis

Exosome biogenesis pathway is initiated by endocytosis of molecular cargo into the cell. Early endosome, the initial vesicle generated by the plasma membrane budding into the cell, is the first stop in the endosomal trafficking pathway; its role is to perform primary sorting and fate determination of the endocytosed cargo [1,2]. There are three paths that the cargo can take from the early endosome. Cargo that needs to be recycled will localize to the peripheral tubular domains of the endosomes, from where it will separate to fuse with the Golgi network or the plasma membrane in the recycling endosome. Cargo not destined for recycling will concentrate at the central vacuolar regions of the early endosome and commit to the endosomal maturation pathway, eventually forming the late endosome. Late endosomes will follow one of the two fates: fusion with lysosomes and subsequent degradation or fusion with the plasma membrane and exosome release (Figure 1) [1]. In addition to changes in subcellular localization, endosomal maturation process is accompanied by changes in the endosomal membrane. First, the endosome changes its membrane composition to allow for its downstream mobility and sorting. Sphingomyelin is exchanged for ceramides, and Rab5, a marker of an early endosome which participates in delivery of vesicles toward the cell center, is substituted for Rab11, which plays a role in late endosome trafficking [3,4]. Second, as vesicular maturation is taking place, certain regions of the endosomal membrane start to invaginate and bud away from the cytoplasm into the intraluminal space of the endosome. The generated intraluminal vesicles (ILVs), in which the cargo is now enclosed, lead to the multivesicular appearance of late endosomes, giving them the name “multivesicular bodies” (MVBs) (Figure 1) [5,6]. If an MVB goes down the path of lysosomal fusion, the cargo within the ILVs will be degraded. However, if an MVB fuses with the cell plasma membrane instead, the ILVs will be secreted into the extracellular space, becoming exosomes.

### 1.1. ESCRT-Dependent ILVs Biogenesis

Upon fusion of MVBs and cell membranes, ILVs will be released into the extracellular space becoming exosomes; as such, ILV biogenesis is biogenesis of future exosomes. Budding of endosomal limiting membrane to generate ILVs is regulated by many pathways and molecules, which can be broken down into two general categories: ESCRT-dependent and ESCRT-independent [7,8,9] (Figure 2). ESCRT (endosomal sorting complex required for transport*)* is a multiprotein machinery that coordinates molecular binding and membrane deformation events that result in biogenesis of and cargo recruitment to ILVs. ESCRTs consist of class E vacuolar protein sorting (Vps) proteins, which assemble into four distinct complexes: ESCRT 0, I, II, III. ESCRT 0 is composed of Vps 27/Hrs (yeast/human orthologues) and Hse1/STAM; ESCRT I consists of Vps 23/TSG101, Vps 28, Vps 37, Mvb12; ESCRT II–Vps 22/EAP30, Vps 25/EAP25, Vps 36/EAP45; ESCRT III–Vps 20/CHMP6, SNF7/CHMP4, Vps 24/CHMP2, and Vps 2/CHMP3 [10]. Vps4 is an AAA ATPse complex that is tightly associated with ESCRTs and is considered to be a part of the machinery. The main function of ESCRT 0-IIs is sorting cargo into functional microdomains on the endosomal membranes; these microdomains serve as organizing centers for assembly of molecular machinery that facilitates signal transduction. ESCRT III is responsible for budding and scission of these domains to produce ILVs, and Vps4 facilitates dissociation of ESCRT III after membrane scission is complete.

The assembly of ESCRT machinery on the endosomal membrane begins when Vps27/Hrs of ESCRT 0 is localized to an early endosome through its FYVE and coiled coil domains binding to early endosomal transmembrane proteins [11]. Vps27/Hrs then recruits ESCRT I by binding its Vps 23/TSG101; Vps 28 of ESCRT I then binds Vps36/EAP45 of ESCRT II [12,13]. Finally, assembly of ESCRT III takes place when ESCRT II binds Vps 20/CHMP6 of ESCRT III, an event which facilitates polymerization and activation of ESCRT III unit SNF7/CHMP4 and subsequent recruitment of ESCRT III units Vps 24/CHMP2 and Vps 2/CHMP3 to the endosomal membrane [14]. SNF7/CHMP4 acts as the main driver of ESCRT III’s role in membrane deformation leading to its inward budding and ILVs generation. Polymerization of SNF7/CHMP4s forms filament spirals which store potential energy; as the spirals elastically compress, this energy is released to create negative curvature in the membrane [15]. After membrane fission is complete, ESCRT III is disassembled when its polymers translocate through the central pore of AAA ATPase Vps4 [16,17]. In addition to its role in ESCRT III disassociation, Vps4 has been shown to participate in the membrane remodeling process itself through interaction with ESCRT III to stabilize necks of the growing ILVs [18].

In addition to the four ESCRTs, there are many other molecular players that participate in ILV generation within the ESCRT-dependent pathway (Figure 2). For instance, ESCRT-associated protein ALIX is intricately involved in the process. In yeast it has been shown that aside from the classical ESCRT 0—ESCRT I—ESCRT II—ESCRT III pathway, ESCRT III can be recruited by an alternative pathway of ESCRT 0—Bro1/ALIX—SNF7/CHMP4 [19]. In mammals, ESCRT III is recruited fully independently of upstream ESCRT complexes in sorting and delivering tetraspanins to exosomes by ALIX in the presence of lysobisphosphatidic acid (LBPA) [20]. Bro1 (yeast homologue of mammalian ALIX) interacts with ESCRT III, Vps4, and ubiquitin conjugated to the cargo protein to create a Bro1—Vps4—ESCRT III axis, which plays a role in ILV formation [21]. Furthermore, ALIX can associate with syndecans, a class of transmembrane proteins, through a scaffolding protein syntenin to participate in the membrane budding steps of ILV biogenesis [22]. In EGF-stimulated cells, annexin-1, a substrate for EGFR tyrosine kinase, as well as phosphatidylinositol (PI) 3′-kinase, participates in the process of ILV generation [23,24]. In Charcot–Marie–Tooth (CMT) disease, mutation of small integral membrane protein of the lysosome/late endosome (SIMPLE) results in decreased exosome biogenesis on endosomal membrane [25]. Just like proteins, membrane lipids actively participate in membrane budding. For instance, Pi3P, a phospholipid specific to endosomal membranes, binds to and helps with recruitment of early ESCRT protein Vps27/Hrs [26]. More generally, the shape and curvature of the endosomal lipid membrane has been implicated in ILV generation, since ESCRT complexes preferentially assemble on highly curved surfaces [6,27].

### 1.2. ESCRT-Independent ILVs Biogenesis

While classically ESCRTs are involved in ILV generation, some proteins and lipids allow for the process to take place in an ESCRT-independent manner (Figure 2). For instance, tetraspanins participate in various steps of exosome biogenesis pathway, such as directing cargo toward MVBs, compartmentalizing endosomal membrane into functional domains (tetraspanin enriched domains (TEMs)), and increasing exosome secretion of certain compounds [28,29]. Tetraspanin CD63 in particular has gotten significant attention for its role in vesicular transport and tumor signaling [30]. Gi-coupled S1P1 receptors have also been implicated in MVB maturation, although the exact mechanism remains elusive [31]. Ceramides, membrane sphingolipids, are greatly involved in ESCRT-independent membrane deformation. They organize the plasma membrane into nanoscale assemblies of sphingolipids, cholesterol, and proteins, termed lipid raft microdomains, which induce spontaneous negative curvature of the membrane and lead to IVL generation in the absence of ESCRT III [32]. Recently, activated Rab31 GTPase was identified as the trigger for membrane budding within these microdomains [33]. The ceramide transfer protein (CERT) plays a central role in ceramide mediated exosome biogenesis and secretion, since it facilitates transfer of ceramides onto early and late endosomes from the Golgi and ER networks [34].

### 1.3. ILVs Cargo Sorting Machinery

Before these processes of inward membrane budding on endosomes can take place, the appropriate cargo needs to be recruited to the vesicles. Since the cargo consists of different classes of molecules, such as proteins, nucleic acids, and lipids, the mechanisms for their endosomal sorting are also distinct (Figure 3). The signal for proteins to join the endosomal pathway is monoubiquitination (unlike polyubiquitination that targets cargo for proteosomal degradation), which allows the cargo to become recognized by and bound to ubiquitin-interacting-motif (UIM) domain of ESCRT 0 Vps27/Hrs protein [35,36]. By binding to Vps 27/Hrs, the ubiquitinated cargo is concentrated in clathrin-rich microdomains of the endosomes, regions which will bud away from the cytoplasm to generate ILVs [37,38]. In addition to ESCRT 0, ESCRT I and II also contain ubiquitination recognition motifs that help to direct and spatially organize the cargo [39]. Before being enclosed in ILVs, ubiquitin is removed from the cargo proteins by deubiquitinating enzymes. One such enzyme is ubiquitin thiolesterase Doa4, which gets localized to endosomal membranes by binding to ALIX/Bro1 [40]. In tumor cells, in addition to ubiquitination, phosphorylation of Vps27/Hrs by extracellular signal-regulated kinases (ERKs) favors delivery of PD-L1 protein to exosomes [41]. Not all protein sorting requires ubiquitination. For instance, recruitment of integral membrane protein Cvt17/Aut5p and Sna3p protein in yeast is ubiquitination independent [42,43]. Similarly, unubiquitinated IL-2Rb is sorted to the endosomal pathway by binding to the non-UIM domain of Vps 27/Hrs [44]. Other proteins, such as GPCRs, can be recruited by binding to ALIX instead of Vps 27/Hrs [45]. Interestingly, ESCRT 0 has a role in cargo sorting beyond Vps 27/Hrs [46]. Just like some proteins do not require the ubiquitin tag to be directed to the endosomal pathway, sorting of other cargo is independent of the ESCRT machinery. Proteins with KFERQ motif require LAMP2A and a molecular chaperone HSC70, in addition to Alix, CD63, Syntenin-1 and RAB31, without the need for ESCRT complexes [46]. Recently, there has been an increasing body of evidence on interconnections between exosome secretory and autophagy pathways in the cells. Cargo sorting is one process in which autophagy machinery is involved: LC3-II located on the MVB membrane participates in recruitment and sorting of RNA-binding proteins, such as heterogeneous nuclear ribonucleoprotein K (HNRNPK) and scaffold-attachment factor B (SAFB), to ILVs [47].

While posttranscriptional modification by ubiquitination is known to be a common tag to direct proteins to exosomes, loading of non-protein cargo is understood less. miRNA can be directed to exosomes by binding to a selectively sumoylated (conjugated with small ubiquitin-related modifier) heterogeneous nuclear ribonucleoprotein A2B1 (hnRNPA2B1) expressed in the exosomal membranes [48]. In KRAS colorectal cancer cells, KRAS and Ago2 participate in targeting miRNAs to exosomes [49,50]. Other proteins have been implicated in RNA loading into exosomes, such as major vault protein (MVP) for miR-193a in colon carcinoma cell line, HuR for miR-122 in human hepatic cells, Arc protein for mRNA in neurons, etc. [51,52,53]. The abovementioned proteins, as well as many others, have been grouped into the RNA-binding protein (RBPs) class of exosomal RNA sorting [54]. Whether specific mechanisms for DNA sorting into exosomes exist is still poorly understood, since several studies observed complete genome sequences of the parent cell in their exosomes [55].

### 1.4. Commitment of ILVs to the Exosome Pathway

After cargo sorting and ILV generation processes are complete, MVBs face two choices: to fuse with lysosomes for cargo degradation or with the plasma membrane for release of ILVs as exosomes. Although, some recent discoveries have proposed additional fates for ILVs: ILVs can undergo retrofusion with the MVB membrane instead of becoming exosomes or be secreted from the cells after lysosomal fusion by the process of lysosomal exocytosis [56,57]. While the specifics of what commits MVBs to the classical lysosomal or cell membrane pathway have not been fully elucidated, a variety of factors has been shown to play a role. For instance, in B-lymphocytes, only MVBs with high cholesterol content are able to fuse with the plasma membrane and release their ILVs as exosomes [58]. On the other hand, ISGylation of MVB protein Vps 23/TSG101 has been shown to favor MVB fusion with lysosomes and direct them away from the secretory pathway [59]. Vps23/TSG101 is also a target of ubiquitination by Mahogunin Ring Finger-1 (MGRN1) protein, which favors fusion of late endosomes with lysosomes [60]. Several Rab GTPases, a group of small GTPases that coordinate events of vesicular traffic, participate in determination of MVB’s fate. Rab7, a GTPase that regulates transport and fusion of late endosomes with lysosomes, promotes degradation of MVBs, thereby decreasing exosome secretion [61]. The mechanism lies in creation of NEDD8-Coro1a complex on the MVB membrane, which in turn recruits Rab7 [62]. On the other hand, Rab31 counteracts the action of Rab7 and promotes exosome secretion by recruiting GTPase-activating protein TBC1D2B, which subsequently deactivates Rab7 [33].

Autophagy proteins further play a role in guiding MVB traffic. One example is involvement of Atg5, an autophagosome protein, in MVB fusion with the plasma membrane through regulating pH of the late endosomes [63]. Moreover, autophagosomes can either fuse with lysosomes for degradation of their contents or with MVBs for secretion of their cargo into the extracellular space [64]. For this reason, factors affecting the rate of secretory autophagy in turn affect the rate of exosome release. For instance, phosphorylation of autophagosome proteins by ATM, a player in the DNA-repair machinery, in cancer-associated fibroblasts (CAFs) under conditions of hypoxia favors autophagosome fusion with MVBs over lysosomes [54].

Once the MVBs are committed to the exosome pathway, they are transported from the center of the cell to its periphery. There are a few key players in this vesicular traffic system: actin filaments and microtubules, along which the vesicles move, motor proteins, such as kinesin and dynein, which directly facilitate the movement, and Rab family of small GTPases which recruit and activate the motor proteins [65,66]. One example of a motor protein that promotes exosome secretion through controlling vesicular transport is cortactin, an actin regulatory protein; in tumor cells, its knockdown leads to a decrease of exosome secretion, and its overexpression produces the opposite effect [67]. The types of Rabs which participate in trafficking MVBs to the cell membrane are highly variable between different organisms and cell types. For instance, in human leukemia K562 and drosophila S2 cells, but not in HeLa cells, downregulation of functional Rab11 leads to decrease in exosomal secretion [68,69,70]. Others GTPases, such as Rab2b, Rab5a, Rab9a, Rab27, and RAL-1 have also been shown to participate in secretion of exosomes [70]. Alix and clathrin are likewise involved in directing MVBs to the plasma membrane, most likely by interacting with plasma membrane-associated actin networks [71,72].

## 2. Exosome Release

After the MVBs get delivered to the plasma membrane of the cell, they follow a general scheme for vesicular docking and fusion onto a cell membrane, with main players consisting of v-SNAREs (on vesicles), t-SNAREs (on target membranes), Rab GTPases, tethers, and additional proteins (Figure 4) [73]. Typically, multiple binding complexes of one v-SNARE and three t-SNAREs occur between the merging membranes. Rabs, such as Rab27a, Rab27b, and Rab35, ensure proper membrane targeting by recruiting specific tethers to bind to the SNARE proteins, as well as take part in vesicular docking at the cell membrane [70,73,74]. SNARE proteins, such as VAMPs (v-SNAREs), syntaxins (t-SNAREs), and SNAPs (t-SNAREs), play a crucial role in the secretory process by facilitating the fusion of the endosomal and plasma membranes. For instance, VAMP7 localized to late endosomes forms VAMP7/syntaxin 1/SNAP-25 and VAMP7/syntaxin 3/SNAP-23 complexes, allowing for the abovementioned fusion to occur [75,76]. In an Alzheimer’s disease model in neurons, another late endosomal v-SNARE VAMP8 participates in fusion of tau-carrying vesicles with the cellular membrane [77]. Syntaxin 4 is involved in Hepatitis C virus (HCV) spread by facilitating fusion of virus-carrying MVBs with the membranes of infected cells, leading to HCV’s release in exosomes [78]. Furthermore, in Parkinson’s disease models, increased concentration of α-syn correlated with decreased interaction between syntaxin 4 and VAMP2, leading to decreased exosome secretion [79]. Knockdown of another t-SNARE, syntaxin 6, in prostate cancer cells significantly decreased exosome production and reduced drug resistance conferred by this secretory mechanism [80]. While involvement of many SNARE proteins is cell type-specific, it has become apparent that VAMP7 and SNAP-23 are ubiquitously central to the membrane fusion process [75].

Many regulatory mechanisms of exosome secretion occur at the plasma membrane through the SNARE complex. One group of such regulations is post-translational modifications of SNARE proteins, such as O-GlcNAcylation and phosphorylation. Reduced O-GlcNAcylation of SNAP-23 promotes its interaction with syntaxin 4 and VAMP 8, leading to increased exosome secretion; phosphorylation of a SNAP-23 by an activated histamine H1 receptor in HeLa cells produces a similar effect [81,82]. Another study in cancer cells showed that phosphorylation of SNAP-23 at Ser95 by PKM2 upregulated exosome release [83]. In addition to posttranslational modifications, RNAs have been implicated in regulation of SNARE complex proteins and, subsequently, of exosome release. In non-small cell lung cancer models, miR-134 and miR-135b microRNAs inhibit SNARE protein YKT6 and thereby reduce exosome secretion [84]. In pancreatic cancer cells, long non-coding RNA (lncRNA) PVT-1 plays a role in MVB fusion with the plasma membrane by regulating colocalization of YKT6 and VAMP3, as well as palmitoylation of YKT6 [85]. Furthermore, lncRNA HOTAIR affects colocalization of SNAP-23 with VAMP3, as well as induces phosphorylation of SNAP-23 [86].

## 3. Exosome Uptake

After MVB and cell membranes fuse, ILVs are secreted into the extracellular space as exosomes. Currently, the mechanisms and players of extracellular vesicle (EV) targeting are not fully understood, and there is still a question of how much of the exosome delivery is stochastic rather than destination-specific [87]. However, it is known that once an exosome reaches its target cell, it can affect that cell in one of three ways: directly interact with its plasma membrane receptors through exosomal surface proteins, fuse with its membrane, or undergo endocytosis (phagocytosis, micropinocytosis, lipid raft- or clathrin- or caveolin-mediated endocytosis) (Figure 5) [88,89,90,91,92,93]. Exosome surface molecules, such as tetraspanins, immunoglobulins, proteoglycans, and lectin receptors are involved in exosomes binding to target cells through mechanisms that are largely unknown [93,94,95]. Exosomal ligands that are currently of the most therapeutic interest are PD-L1, TNF, FasL, and TRAIL, since their receptors are located on the surfaces of tumor cells, thereby making them potential targets for cancer therapies.

When an exosome directly fuses with the cell membrane, its contents are released into the cytoplasm and its journey is complete. While this pathway is the most efficient one for cargo delivery into the cell, evidence suggests that the dominating mechanism of exosome uptake is endocytosis: an intact exosome is engulfed and bound by the plasma membrane, subsequently joining the endosomal system [87]. In this case, for the exosome cargo to produce its effects, it needs to escape from vesicles into the cytoplasm (“endosomal escape”); if it remains within the endosomal pathway, it will be degraded by a lysosome, recycled within the cell, or secreted to the extracellular space without affecting the function of the target cell [96,97,98]. A few proposed mechanisms for cargo to reach the cytoplasm are fusion with endosomes in a pH dependent manner and permeabilization of and subsequent escape from endolysosomes [99,100]. Once in the cytoplasm, the cargo can carry out its designated function. There is still much to be understood about the mechanisms of exosome cargo release into the cytoplasm, since poor predictability of the process across different cell types is currently one of the main challenges in utilizing the extracellular vesicle system for clinical applications [98,101,102].

Since exosomes travel in the blood, before they reach their target organs, they must cross the endothelial layer of the vasculature. While much remains to be understood about the mechanisms of this process, recent studies on EV passage through the blood–brain barrier (BBB) have suggested transcytosis as the most plausible mechanism of exosome transport across the endothelium [103]. Breast-cancer derived EVs have been suggested to enter endothelial cells via clathrin-mediated endocytosis; they were then sorted by Rab11 for exocytosis at the basolateral membrane and were finally secreted from the cell through interactions of a v-SNARE VAMP-3 on the EVs with membrane associated t-SNAREs SNAP23 and syntaxin 4 [104]. In a separate study, heparan sulfate proteoglycans on the surface of endothelial cells were shown to be involved in exosome endocytosis; however, it is not clear whether they participate in exosome internalization or attachment [105]. Adsorptive transcytosis, a mechanism that facilitates transport through the cell by interactions of positive and negative molecular charges, has also been suggested [106].

## 4. Limitations and Challenges in Exosome Research

Existence of many imperfect exosome isolation and purification protocols has been one of the greatest limiting experimental factors in exosome research. Size exclusion and affinity-based chromatography, as well as ultracentrifugation are currently the mainstay approaches in the field [107]. However, each method has its own caveats—ultracentrifugation does not rid the samples of other components of similar sizes and densities, and affinity-based methods possess limited purification power, as no surface marker is exclusive to exosomes [108]. To get purer exosome population, researchers are encouraged to combine several methods in their protocols. For example, several studies reported the presence of mitochondria proteins and nucleic acids in their isolated exosomes. However, by combining ultracentrifugation with Opti-Prep based step gradient density column, one group was able to separate out a similar in size, yet characteristically distinct, population of sEVs of mitochondrial origin, mitovesicles, and thereby obtain a more pure exosome fraction [109,110,111,112,113,114].

Another significant challenge when working with exosomes is their storage conditions. While freezing samples to temperatures below biochemical reactivity is the standard approach for sample storage in research, scientists working with exosomes must get more creative, as freeze–thaw cycles may cause formation of exosomal aggregates. Freeze-drying or adding cryo-stabilizers could improve exosome quality for storage and transportation, but little is known about significance or effectiveness of using these methods [115].

In addition, there is a nomenclature discrepancy between research groups is “exosomes” and “small extracellular vesicles” (sEVs). While all exosomes are sEVs, not all sEVs are exosomes [114]. Since the two can be of similar sizes and since exosomes possess a great variety of surface markers, it can be difficult to draw definitive experimental conclusions about sEVs vs. exosomes. In order to maximize standardization of methods across the field, International Society for Extracellular Vesicles (ISEV) has already released two versions of Minimal Information for Studies of Extracellular Vesicles (MINEV), one in 2014, and a revision in 2018. MINEV is a set of guidelines produced by the leaders in the field with the goal of introducing standard methods for EV study to increase rigor and reproducibility of research conducted by different institutions. Regarding the EV nomenclature, the guidelines generally suggest the use of the term “extracellular vesicles” with specifications of size and/or surface markers. On the topic of isolation and purification methods the consensus is less clear, and the recommendations are limited to suggesting researchers to choose methods appropriate for their respectful downstream analysis. Currently, the latest edition of MINEV is in the works, leaving us hopeful for improved protocol clarity in the near future.

## 5. Conclusions and Future Prospects

Over the span of the past few decades, the field of exosomes has greatly expanded, changing their status from cellular garbage disposals to promising diagnostic and therapeutic tools. In the past 10 years, there has been a steady increase in the number of research articles and review papers published per year on exosomes, a trend which suggests an increasing interest in advances, as well as the need for systematic summaries of existing knowledge in the field. In our review, we bring together well-established understanding of exosome machinery and overview of discoveries from the past two years, focusing on exosome biogenesis, release, and uptake. Most up-to-date knowledge of these pathways combined with development of novel techniques is crucial for advancing the exosome field in the direction of clinical applications. In the upcoming years, with increasing availability of the exosome reporter mice (ex. CD63-GFP mice [116] and Cre-inducible His-tagged CD9/TurboGFP reporter mice [117], Jackson Lab), tissue specific function of exosomes is expected to widely explored through in vivo gain- and loss-of-function experiments. Another technique that is expected to gain popularity is utilization of novel live cell reporters of exosome secretion and uptake (ex. pHluo_M153R-CD63) to study the complete life cycle of the vesicles in question [118]. Such genetic tools, as well as high sensitivity exosome isolation and purification protocols, will allow this evolving field to create a significant impact on healthcare in the future.

## Figures and Tables

**Figure 1 ijms-24-01337-f001:**
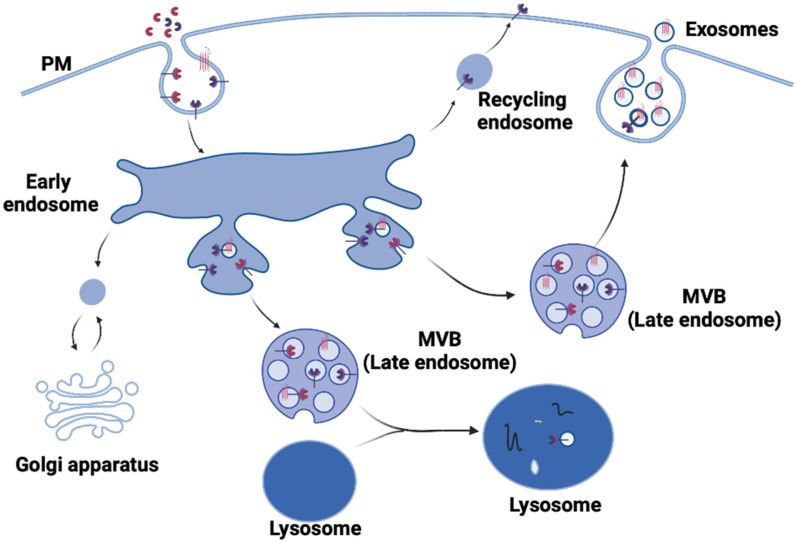
Schematic of the exosome machinery. Cargos are sorted by endocytosis to the early endosomes. Early endosomes are committed to the endosomal maturation pathway, which results in multivesicular appearance of the late endosomes (MVB). Finally, MVBs fuse with the plasma membrane (PM) to release exosomes. Created with BioRender.com.

**Figure 2 ijms-24-01337-f002:**
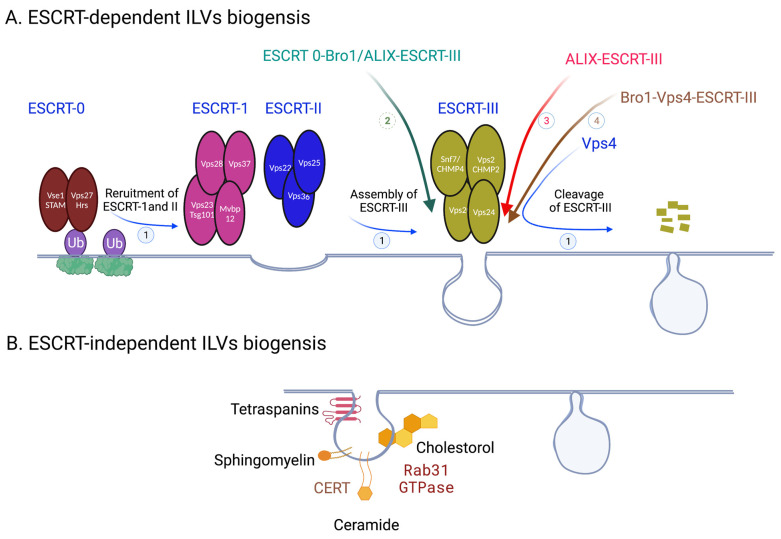
Exosome biogenesis. (**A**) ESCRT-dependent ILVs biogenesis. 1. Classic pathways from ESCRT 0-III-Vps4. 2. ESCRT 0-Bro1/ALIX-ESCRT III. 3. ALIX-ESCRT III. 4. Bro1-Vps4-ESCRT III. (**B**) ESCRT-independent ILVs biogenesis. Created with BioRender.com.

**Figure 3 ijms-24-01337-f003:**
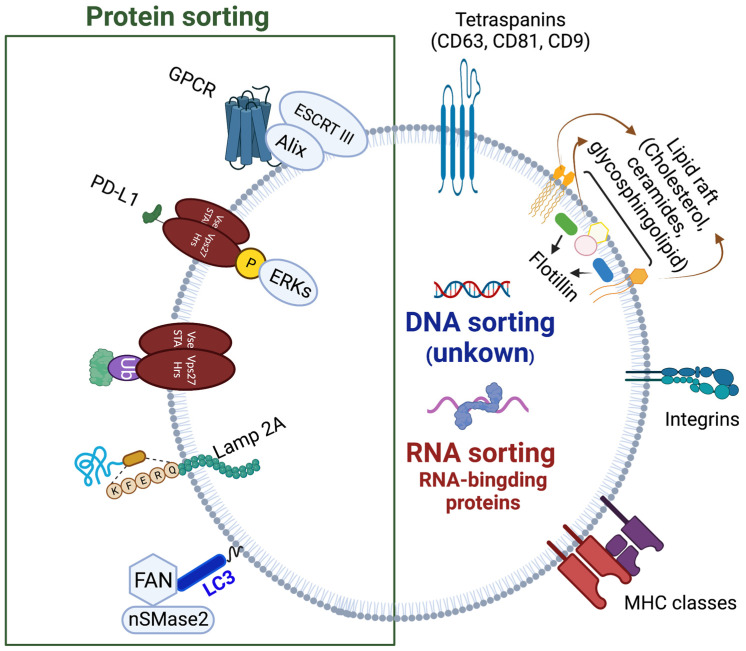
Cargo sorting into exosomes. The known pathways of protein, RNA, and DNA sorting are depicted. Exosomal membrane proteins including tetraspanins, flotillin, integrins, MHCs, and ESCRT are labeled. Created with BioRender.com.

**Figure 4 ijms-24-01337-f004:**
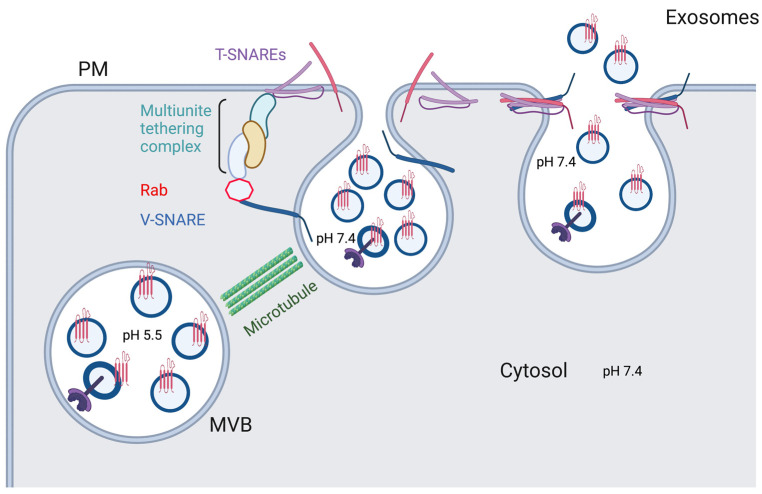
Exosome secretion. With the help of microtubules, Rabs, and tethering factors, one vesicle SNARE (V-SNARE) forms four-helix bundles with two target SNAREs (T-SNARE) to drive the fusion between MVB with plasma membranes to secret exosomes. Created with BioRender.com.

**Figure 5 ijms-24-01337-f005:**
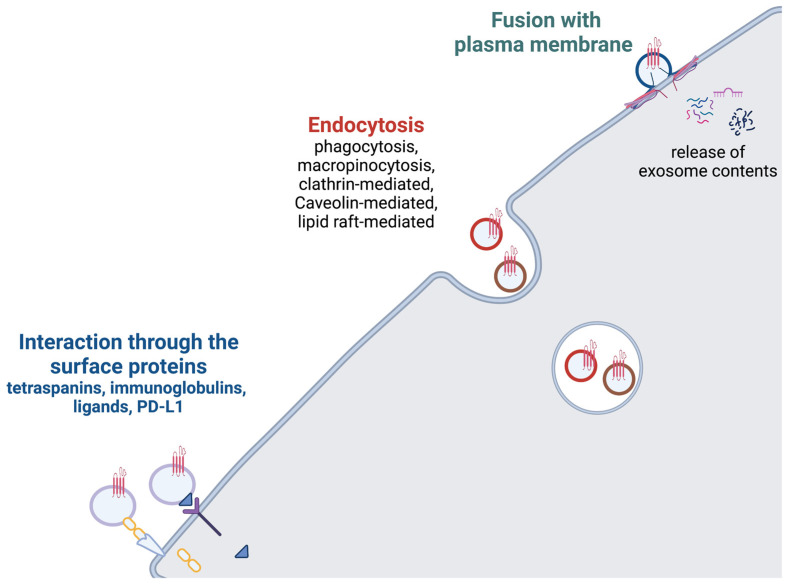
Exosome uptake. Exosome uptake occurs via one of three pathways: (1) Fusion with the plasma membrane to release exosome contents; (2) Endocytosis; (3) Direct interaction with surface receptors. Created with BioRender.com.

## Data Availability

No new data were created or analyzed in this study. Data sharing is not applicable to this article.

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
