# Peer review of "The Machinery of Exosomes: Biogenesis, Release, and Uptake"

_ijms, 2023, doi:10.3390/ijms24021337_

Round 1
Reviewer 1 Report
The topic of this review is highly relevant and important. The references included in the article are informative. The manuscript is relatively well-written. This article, however, suffers from two major defects:
1. The lack of novelty
The topic of this review article has been extensively studied with a great number of publications. The authors have failed to present the novel approaches and findings from their current study compared to those of previous studies.
2. The lack of analytic assessment
The presentation of the references is primarily descriptive in nature.
I would like to strongly encourage the authors to select a more specific field to dig deeper, and to present more updated information.
Author Response
Comments and Suggestions for Authors
The topic of this review is highly relevant and important. The references included in the article are informative. The manuscript is relatively well-written. This article, however, suffers from two major defects:
- The lack of novelty
The topic of this review article has been extensively studied with a great number of publications. The authors have failed to present the novel approaches and findings from their current study compared to those of previous studies.
We added the novel approaches and findings from current studies compared to those of previous studies in section 4 “Limitations and Challenges in Exosome Research” as well as in the section 5 “Conclusion and Future Prospects”.
- The lack of analytic assessment
The presentation of the references is primarily descriptive in nature.
I would like to strongly encourage the authors to select a more specific field to dig deeper, and to present more updated information.
There has been an increasing number of reviews in the field of exosomes or extracellular vesicles (EV) published every year over the past 10 years, such trend suggests that summarizing the latest advances in the field is in high demand and is crucial for further research to take place. Our review not only summarizes the well-established basics of exosome biogenesis machinery, but also provides information on the most recent advances in the field, thereby making it useful for a wide scientific audience.
It is true that the main body of our review is mostly descriptive since the intent of our work was to summarize and describe the current state of knowledge of the exosome machinery. Although, in the section 4 “Limitations and Challenges in Exosome Research” as well as in the section 5 “Conclusion and Future Prospects” we have added our analytical input.
Reviewer 2 Report
Sofia Krylova and Daorong Feng wrote an interesting review on biogenesis, release and uptake of Exosomes. This is worth for researchers working on exosomes/extracellular vesicles.
Minor comments
1. Abstract should not contain any reference.
2. Section 1 (The overview of the Exosome Biogenesis Pathway): I would suggest to start with extracellular vesicles (EV), because exosome is a type of EVs. Before exosomes, highlighting EVs is very important for the readers.
3. Section 4 can be expanded, I would suggest to include the limitations and challenges in exosome research. Because, it is important to have knowledge about this, while designing the experiments. Also, I would suggest to cite the paper Karn et al., 2021 (Extracellular Vesicle-Based Therapy for COVID-19: Promises, Challenges and Future Prospects ; https://doi.org/10.3390/biomedicines9101373) for this.
4. Section can be written as Conclusion and future prospects. Conclusion is very important for any paper.
Overall, its nicely presented.
Author Response
Sofia Krylova and Daorong Feng wrote an interesting review on biogenesis, release and uptake of Exosomes. This is worth for researchers working on exosomes/extracellular vesicles.
Minor comments
- Abstract should not contain any reference.
All references in Abstract have been deleted.
- Section 1 (The overview of the Exosome Biogenesis Pathway): I would suggest to start with extracellular vesicles (EV), because exosome is a type of EVs. Before exosomes, highlighting EVs is very important for the readers.
To match the title, we changed “The overview of the Exosome Biogenesis Pathway” to “Exosome Biogenesis”.
- Section 4 can be expanded, I would suggest to include the limitations and challenges in exosome research. Because, it is important to have knowledge about this, while designing the experiments. Also, I would suggest to cite the paper Karn et al., 2021 (Extracellular Vesicle-Based Therapy for COVID-19: Promises, Challenges and Future Prospects ; https://doi.org/10.3390/biomedicines9101373) for this.
Very good point, we expanded the section 4 and added the reference.
- Section can be written as Conclusion and future prospects. Conclusion is very important for any paper.
Overall, its nicely presented.
Thanks, we added conclusions in section 5.
Reviewer 3 Report
Dear Editors and Authors,
Thanks for the opportunity to read the paper entitled “The machinery of exosomes: biogenesis, release and uptake” written by Sofia Krylova and Daorong Feng. I enjoyed the reading and I agreed with the authors that a good understanding of the mechanisms involved in the biogenesis of exosomes and other EVs would benefit their successful clinical practice. But I have a few concerns:
Major concerns
1. Line 60-61, please add van Niel’s papers for ESCRT-independent pathways of protein sorting in MVB.
2. Line 65-74, as a review article written by specialists in the field of metabolism and diabetes, I strongly refer the authors to the recent excellent reviews (doi: 10.15252/embj.201592484, doi:10.1038/nrm.2016.121 and doi: 10.1146/annurev-cellbio-100616-060600), and encourage to draw a table of composition of the ESCRT complexes and cite original articles, instead of citing an article focusing on plants that was written over a decade ago.
3. Line 123-124,”Tetraspanin CD63….”, again, please cite the reference mentioned in the first concern.
4. Regarding the section on caution in exosome studies (line 318-332), in addition to definitions and protocols, the recently published minimal requirements for the study of extracellular vesicles (DOI 10.1002/jev2.12182) should be considered by referring to and discussed.
Minor concerns
1. Line 47-48, “which plays a role in late endo-47 some trafficking, and [12, 13].” sentence to be finished.
Author Response
Comments and Suggestions for Authors
Dear Editors and Authors,
Thanks for the opportunity to read the paper entitled “The machinery of exosomes: biogenesis, release and uptake” written by Sofia Krylova and Daorong Feng. I enjoyed the reading and I agreed with the authors that a good understanding of the mechanisms involved in the biogenesis of exosomes and other EVs would benefit their successful clinical practice. But I have a few concerns:
Major concerns
- Line 60-61, please add van Niel’s papers for ESCRT-independent pathways of protein sorting in MVB.
Thanks, we added.
- Line 65-74, as a review article written by specialists in the field of metabolism and diabetes, I strongly refer the authors to the recent excellent reviews (doi:10.15252/embj.201592484, doi:1038/nrm.2016.121and doi: 10.1146/annurev-cellbio-100616-060600), and encourage to draw a table of composition of the ESCRT complexes and cite original articles, instead of citing an article focusing on plants that was written over a decade ago.
Thanks. The references have been added according to the reviewer. The figure 2 already showed the composition of the ESCRT complexes and will not draw a table because of the limitation of the length of the paper.
- Line 123-124,”Tetraspanin CD63….”, again, please cite the reference mentioned in the first concern.
Thanks, the reference has been added.
- Regarding the section on caution in exosome studies (line 318-332), in addition to definitions and protocols, the recently published minimal requirements for the study of extracellular vesicles (DOI 10.1002/jev2.12182) should be considered by referring to and discussed.
Thanks, the discussion about MINEV has been added in section 4.
Minor concerns
- Line 47-48, “which plays a role in late endo-47 some trafficking, and [12, 13].” sentence to be finished.
Thanks, it is fixed.
Reviewer 4 Report
This is a well-organized manuscript with clear figures. The content covers the newest knowledge regarding the biogenesis, release, and uptake of exosomes, and underlying mechanisms. I highly appreciate the contribution of the authors to provide a precious review article for the readers. There is no major question and some few minor points are listed as follows:
1. To avoid confusion for readers, authors are encouraged to move the paragraph of the nomenclature discrepancy of exosomes to the first section "Overview of the Exosome Biogenesis Pathway".
2. The form of reference is varied at the end of the section "Overview of the Exosome Biogenesis Pathway". Please recheck it.
Author Response
Comments and Suggestions for Authors
This is a well-organized manuscript with clear figures. The content covers the newest knowledge regarding the biogenesis, release, and uptake of exosomes, and underlying mechanisms. I highly appreciate the contribution of the authors to provide a precious review article for the readers. There is no major question and some few minor points are listed as follows:
- To avoid confusion for readers, authors are encouraged to move the paragraph of the nomenclature discrepancy of exosomes to the first section "Overview of the Exosome Biogenesis Pathway".
In the abstract we have established exactly what kinds of vesicles we refer to in our review, so we don’t foresee confusion for our readers on that front. The discrepancies between various research groups and how those discrepancies can complicate the study of exosomes have been discussed in the section 4 – “Limitations and Challenges in Exosome Research”.
- The form of reference is varied at the end of the section "Overview of the Exosome Biogenesis Pathway". Please recheck it.
Thanks, it is fixed.
Round 2
Reviewer 1 Report
The revised manuscript should be properly published in IJMS